# Broadband Time-Delay and Chirp Compensator for X-ray Pulses

**Christoph Braig** [ID] **and Alexei Erko *** [ID]

Institute of Applied Photonics e.V., Rudower Chaussee 29/31, 12489 Berlin, Germany; braig@iap-adlershof.de
* Correspondence: erko@iap-adlershof.de; Tel.: +49-30-6392-6586

**Abstract:** A new type of aberration-corrected time-delay compensating monochromator (TDCM) for soft X-rays is presented. Composed of two identical reflection zone plates (RZPs) on spherical substrates and an intermediate flat mirror for band-pass selection, the TDCM can operate in a wide energy range of about $\pm 20\%$ around the design energy of 410 eV. Assuming a source size of 50 µm and an angular acceptance of 1 mrad, the spectral resolving power may reach $6 \times 10^2$, at a pulse length as short as 4.3 femtoseconds (fs). In the case of µm-sized sources, the resolution can be better than 0.1 eV and the sub-fs regime could become accessible. The overall transmission efficiency varies within (4.2–6.0)% across the energy range (310–510) eV. In the complementary mode, chirped-pulse compression works as well. Depending on the properties of the source, simulations predict an up to 9-fold reduction in pulse duration, whereas $\leq 50\%$ of the peak intensity is maintained.

**Keywords:** XUV/soft X-ray spectroscopy; femtosecond pulses; reflection zone plates; X-ray lasers

## 1. Introduction

In the "water window" from 284 eV to 530 eV, time-resolved soft X-ray spectroscopy probes the dynamics of electrons in atoms, the charge transport, photochemical processes in biomolecules, or the K- and L-edge absorption of, e.g., Ca and Na in their natural environment on an ultra-short, (sub-)femtosecond time scale [1–4]. Typically, wavelength-dispersive gratings are used to monochromatize the X-ray pulses to the desired energy bandwidth [5–8] but their temporal length is stretched due to a pulse-front tilt. The contribution to geometrical pulse elongation, which is proportional to the number of illuminated grating lines [9], can be compensated by a second diffractive optical element. Basic principles have been introduced for plane gratings of a constant period [10] and further analyzed in the extreme ultraviolet (XUV) regime [11]. In spite of their superior efficiency, setups using conical diffraction [12–14] have been restricted, until now, to applications in the XUV, where the grating line density is still low enough to be fabricated. Experimental results on high harmonic generation (HHG) obtained with toroidal gratings were reported [15], the error budget of the alignment was studied [16], and an in-depth investigation of the performance was supported by measurements on HHG pulses, once more [17].

In agreement with our recently reported [18,19] scheme, we suggest combining two identical RZPs placed in opposite diffraction orders, i.e., the $(+1)^{st}$ and $(-1)^{st}$ one, to a time-delay compensating monochromator (TDCM). Nonetheless, the free spectral range of a TDCM constructed with RZPs on plane substrates is limited by their chromatic aberration. Practically, a reasonable resolving power of the TDCM can be achieved only in a range of (3–5)% around the design energy [20]. Furthermore, the critical line density in the lateral direction limits the aperture of "classical" RZPs with elliptical grooves. Finally, the layout from [18,19] can be operated without moving parts [21] only at its design energy and with relatively large sources, due to technical limitations on the minimum slit size in the intermediate Fourier plane.

In this paper, we demonstrate with simulations how performance can be enhanced by a modified design. In Section 2, we review theoretical basics on short X-ray pulses. The optical design is presented in Section 3 and characterized with respect to its optical

behavior in Section 4. The performance of the TDCM is evaluated in Section 5, followed by considerations on chirped-pulse compression in Section 6. We discuss our findings in the context of related work in Section 7 and conclude with a summary of achievements and an outlook in Section 8.

## 2. Femtosecond Pulses in the Time Domain

The electrical field amplitude $\mathcal{E}_p(t)$ of a "typical" X-ray pulse, as considered in this work, can be described [22] in the time domain with variable $t$ and the Planck constant $h$ as

$$\mathcal{E}_p(t) = \sqrt{\mathcal{I}(t)} \, \exp[2\pi i h^{-1} E(t) t] \quad \text{with} \quad \mathcal{I}(t) = \mathcal{I}_c(\tau_{\text{int}}) \exp[-4\ln 2 \, (t/\tau_{\text{int}})^2]. \quad (1)$$

The Gaussian-shaped envelope of the pulse is characterized by the intensity $\mathcal{I}(t)$ with its peak value $\mathcal{I}_c$ and the intrinsic pulse duration $\tau_{\text{int}}$ (FWHM). The instantaneous photon energy $E$ is assumed to vary around a central energy $E_c$ as

$$E(t) = E_c \pm \partial_t E(\tau_{\text{int}}) t \quad \text{with} \quad \partial_t E(\tau_{\text{int}}) \equiv 2\ln 2 \, \pi^{-1} h \, \tau_{\text{int}}^{-2} \sqrt{(\tau_{\text{int}}/\tau_{\text{ftl}})^2 - 1} \quad (2)$$

as the linear up- or down-chirp for the positive or negative sign, respectively, such that $\partial_t E(\tau_{\text{int}}) \to 0$ for $\tau_{\text{int}} \to \tau_{\text{ftl}}$.

### 2.1. Fourier-Transform Limit

The Fourier-transform limit $\tau_{\text{ftl}} = 2\ln 2 \, \pi^{-1} h / \Delta E$ of a Gaussian pulse [18,23] can be derived directly from the "time-energy uncertainty relation" as the duration at the ultimate, physical minimum [24,25] of the time-bandwidth product (TBP) $\Delta E \cdot \tau_{\text{ftl}} \geq 1.825 \, [\text{eV} \cdot \text{fs}]$ for an energy spread $\Delta E$ (FWHM) around $E_c$. It should be noted that the numerical value of the lower bound to the TBP depends on the energy distribution of the pulse—for a $1/\cosh^2(E - E_c)$ spectrum, the minimal TBP would be somewhat ($\approx 30\%$) less than for the Gaussian type, whereas an increase is found in the rectangular case, for instance. $\tau_{\text{ftl}}$ is independent from any geometrical parameters of the source and the optical layout. According to Table 1, a relative bandwidth $\Delta E/E_c$ in the order of at least 1% around the respective central energy $E_c$ is required to access the sub-femtosecond regime.

**Table 1.** Fourier-transform limit across the energy range of interest for a spectral bandwidth of 1.0%.

| $E = (310 \pm 1.55) \, \text{eV}$ | $E = (410 \pm 2.05) \, \text{eV}$ | $E = (510 \pm 2.55) \, \text{eV}$ |
|---|---|---|
| $\tau_{\text{ftl}} = 0.59 \, \text{fs}$ (FWHM) | $\tau_{\text{ftl}} = 0.45 \, \text{fs}$ (FWHM) | $\tau_{\text{ftl}} = 0.36 \, \text{fs}$ (FWHM) |

For the constant ratio $\Delta E/E_c$, about 44 oscillations of the electric field $\mathcal{E}_p(t)$ from Equation (1) with the angular carrier frequency $2\pi E_c/h$ are passed within $\tau_{\text{ftl}}$, independent of the energy $E_c$. The Fourier-transform limit may be reached in the idealized case of a perfectly compressed pulse and, in particular, a quasi point-like source.

### 2.2. Real Pulse Length

In practice, a lower limit to the real pulse length is defined by the convolution of the fundamental duration $\tau_{\text{ftl}}$ with the initial or residual, temporal elongation $\tau_{\text{chirp}}$ of the pulse (Section 6), as it is described by the slope $\partial_t E(\tau_{\text{int}})$ in Equation (2). Besides a geometrical elongation $\tau_{\text{geo}}$ [11,15,18,19], which arises on the one hand from the finite source size ($\tau_{\text{src}}$) and on the other hand due to chromatic wavefront deformation during beam propagation in the wavelength-dispersive optical setup ($\tau_{\text{dsp}}$), and diverse types of aberrations in addition to coma and astigmatism, which are introduced by any optical elements but play a minor role in this concept study, slope errors are especially important. $\tau_{\text{geo}}$ is conveniently calculated by ray tracing (RT) via the optical path difference (OPD). Finally, the total pulse duration $\tau_{\text{tot}}$ can be estimated using Equation (3),

$$\tau_{\text{tot}} = \sqrt{\tau_{\text{int}}^2 + \tau_{\text{geo}}^2} \quad \text{with} \quad \tau_{\text{int}} = \sqrt{\tau_{\text{ftl}}^2 + \tau_{\text{chirp}}^2} \quad \text{and} \quad \tau_{\text{geo}} \geq \sqrt{\tau_{\text{src}}^2 + \tau_{\text{dsp}}^2}. \qquad (3)$$

Whereas the "physical" contribution $\tau_{\text{int}}$ can be considered as an intrinsic property of the electromagnetic field $\mathcal{E}_p(t)$ from Equation (1), the additive blur $\tau_{\text{geo}}$ is introduced by "technical" constraints, especially the source size (Section 4), and the instrumental response function.

## 3. Optical Design

To overcome the drawbacks of the TDCM designed with elliptical RZPs on two plane substrates [18,19], such as a constrained angular acceptance, the narrow spectral range due to chromatic aberration, and a limited resolving power related to the source and slit size, our newly proposed scheme, as shown in Figure 1, is upgraded in a three-fold way [20,26–28]:

- Both RZPs are outlaid for spherical substrates, with the same radius of curvature (ROC). The ROC is optimized with respect to the arm lengths and angles of the RZPs.
- Both RZPs are designed as "hybrid" RZPs [26–28] for meridional (1-D) focusing and sagittal collimation (exactly at the design energy and approximately offside), to relax the grating line density, enabling an enlarged angular acceptance and photon flux.
- The optical components are re-arranged to an "upside down" configuration. To maintain symmetry, a plane, movable mirror (PM) in the shape of a reflective stripe (∼mm) is placed under grazing incidence at the central position between the RZPs.

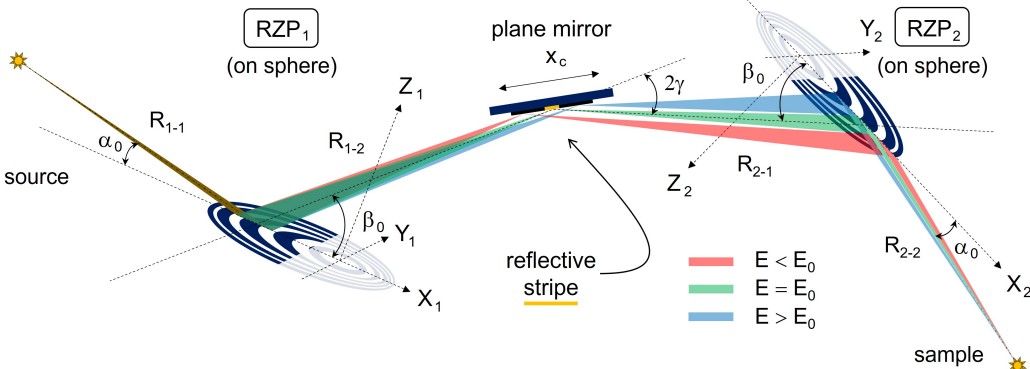

**Figure 1.** Optical layout of the broadband TDCM/chirp compensator, based on two identical RZPs for 1-D focusing (mer.)/collimation (sag.) on spherical substrates. Polychromatic source emission is drawn in brown. The design energy $E_0$ is displayed in green, and red-/blue de-tuned X-ray energies $E$ are addressed by moving the reflective stripe (yellow) of the plane mirror along the main optical axis, to the position $x_c$, which refers to the desired central energy $E_c$ for monochromatization (Section 5). The parameters in both local coordinate systems for $\text{RZP}_{1,2}$ follow the nomenclature in the text.

### 3.1. Reflection Zone Plates and Plane Mirror

The first, "spherical" $\text{RZP}_1$ disperses the incident, polychromatic beam toward the plane mirror (PM), approximately in the "flat field" mode [20]. Being placed in the focal distance of $\text{RZP}_1$, the PM reflects the spatially resolved dispersion pattern toward $\text{RZP}_2$.

With its plane, super-polished surface of negligible roughness on the Ångstrom level, the PM guarantees a nearly aberration- and scatter-free propagation of the beam. Meanwhile, the open aperture of the PM is defined by a reflective stripe of the meridional width $\Delta s$, which enables the selection of the desired energy bandwidth or resolution $\Delta E$. Compared to transmission [18,19], where a slit width in the order of a few $10\,\mu m$ (or even less for best resolution) would be required for monochromatization, a rather wide, mm-sized stripe for use under grazing incidence can be fabricated and handled with reasonable effort at relatively low cost: In a possible process, a series of $\approx$(5–6) prism-shaped wedges could be formed out of one Si block by anisotropic etching. Each wedge has a flat tip of different width $\Delta s$, as controlled by the etching time. Due to protection with a mask, a super-polished

surface quality of the tips is provided and appropriate coating (e.g., Ni) guarantees high reflectivity, as detailed in Section 4. Diffraction at the edges of the stripes is essentially the same as for transmission slits, if the grazing incidence on the stripe is taken into account. With an angular, meridional spread of no more than $\sim$(0.1–0.2) mrad, the relative increase in the pulse duration may be roughly estimated to $\sim$10% or less, and this effect is approximately incorporated in the calculations (Section 5). The three-dimensional (3-D) geometry of the wedges and the weak reflectivity of the surrounding material prevents scattering from the area outside of the reflective stripes. For scanning the energy, one needs to translate the PM in the plane perpendicular to its surface normal and along the main optical axis of the monochromator from the source to the sample. The translational positioning tolerance may be estimated to a factor $1/\sin \gamma$ compared to the slit in transmission, where $\gamma$ denotes the angle of grazing incidence on the PM.

Finally, the second $RZP_2$ focuses the beam—which is divergent in the meridional direction—from the PM onto a sample located in the focal plane of $RZP_2$, simultaneously compensating the time delay introduced by $RZP_1$. As a peculiarity, the spherical substrate allows for a notable spectral tolerance in terms of the almost-fixed position of the output beam on the sample, regardless of the X-ray energy.

In agreement with rules for an optimal performance [23], the RZPs are specified as summarized in Table 2, referring to the notation from Figure 1.

**Table 2.** Design parameters of $RZP_{1,2}$ inside the TDCM, according to Figure 1 with index $(i, j) \, \epsilon \, \{1, 2\}$.

| $\alpha_0 = 2.5°$ | $\beta_0 = 4.0°$ | $E_0 = 410\,\text{eV}$ | $r = 23.8\,\text{m}$ | $490.8\,\text{mm}^{-1}$ | $R_{i-j} = 1.0\,\text{m}$ |
| --- | --- | --- | --- | --- | --- |

Unlike the variable, central photon energy $E_c$ of an interval of width $\Delta E$, as introduced in Section 2, the unchangeable design energy $E_0$ defines the distribution of the Fresnel zones at which the condition for constructive interference of X-rays (with $E_c = E_0$) is matched [9].

The radius of curvature (ROC) of the RZP substrates is represented by $r$. Since the OPD is altered for $r < \infty$ compared to the planar case, the analytical formalism underlying regular RZPs with elliptical Fresnel zones has been generalized to arbitrarily shaped substrates [27]. By means of that method, the correct diffractive structure on a three-dimensional surface is calculated as a kind of "computer-generated hologram", eliminating aberrations at the design energy $E_0$. Using an appropriate technology, such as laser lithography for instance, RZPs on spheres—even with an ROC as short as 4 m [28]—can be realized and applied [20] to soft X-ray spectroscopy over a wide spectral range.

The central grating line density is listed in units of $[\text{mm}^{-1}]$ as the next-to-last value. Entrance and exit arm lengths of the RZPs are equal and identified by symbolic indices $i - j$, like in Figure 1. A relatively low loss in the photon flux is guaranteed for a grazing incidence angle $\gamma = 2.5°$, at which the design energy $E_0$ is reflected on the PM.

*3.2. Angular Misalignment*

In practice, the performance might suffer from misaligned optics. Among all translational and angular degrees of freedom, the critical pitch angles of the RZPs (rotation around axis $Y_{1,2}$) and the PM are considered exemplarily. Figure 2 illustrates the effect of tilted substrates, evaluated in terms of the vertical focus position $z$ as a function of the photon energy $E$.

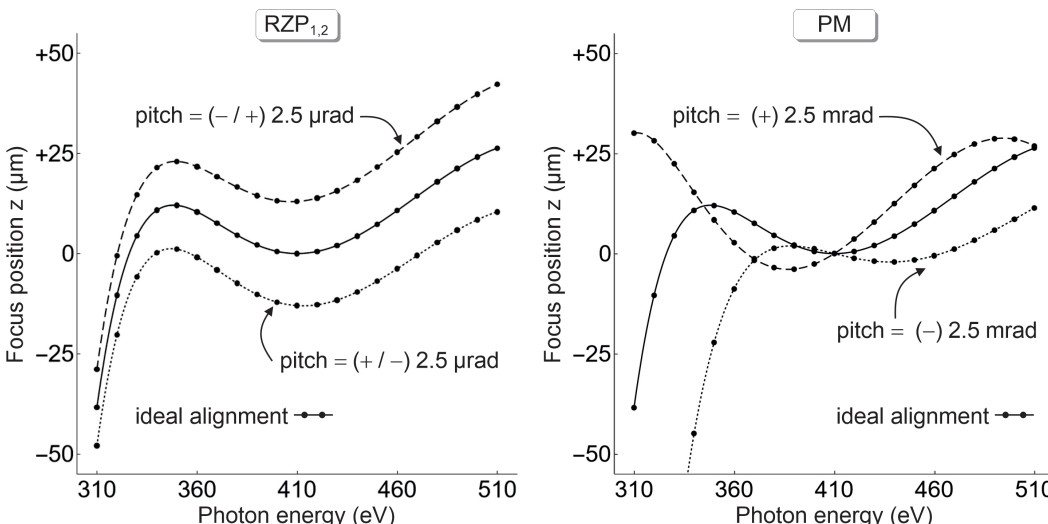

**Figure 2.** Spatial chirp and estimated angular-error budget of the optical components. If both RZPs and the PM are perfectly adjusted, the vertical position $z$ of the focal image of a point source with negligible divergence ("pencil beam") follows the aberration-corrected, achromatic function as described by the solid line. On the left, the case of an erroneous pitch of $\pm 2.5$ µrad, applied in opposite direction to $RZP_1$ and $RZP_2$, leads to displaced curves (dashed/dotted). On the right, a pitch of $\pm 2.5$ mrad is assumed for the PM instead, perturbing the spatial chirp for $E \neq E_0$.

An exactly adjusted PM presumed, $RZP_{1,2}$ may be rotated within a tolerance of about $\pm 2.5$ µrad, to confine the focal displacement to the order of the source size (50 µm). In comparison, not before an up to $10^3 \times$ enlarged, accidental tilt of the PM leads to focal displacements of a similar magnitude, as shown on the right in Figure 2. Interestingly, the inherent achromatic correction of the proposed TDCM, which manifests itself in the slow variation in the spatial chirp $z(E)$ in the focal plane, is relatively stable and, in principle, maintained, even under moderate misalignment. From a technical point of view, the tight requirements, especially with respect to $RZP_{1,2}$, demand precise multi-axis mounting, which can be nonetheless routinely realized in state-of-the-art X-ray optical instrumentation [9].

## 4. Source Size, Pulse Propagation and Transmission Efficiency

In the case of precisely fabricated optical elements and a properly aligned system, the same number of grating lines is covered by the beam with an energy $E_0$ on both RZPs [9], enabling full time-delay compensation [15] for a point source, regardless of its angular divergence $\Delta\vartheta$. At off-design energies (e.g., $E_0 \pm 20$ eV), chromatic aberrations, as represented by $\tau_{geo}$ in Equation (3), stretch the pulse on the sample proportional to $\Delta\vartheta$. Notably, the sagittal component of that divergence dominates the temporal blur by one order of magnitude, compared to the angular acceptance in meridional direction. In accordance with typical beam properties in the soft X-ray regime, we set $\Delta\vartheta = 1$ mrad (FWHM) within this work, such that aberrations contribute no more than $\sim 0.1$ fs to the final pulse duration.

### 4.1. Source Size and Slope Errors

At any X-ray energy, the lateral dimension $\varnothing_{src}$ of a real source lengthens the optical path and thus the photon run-time, too [9]. To evaluate the geometrical, lower bound to the total pulse length $\tau_{tot}$ in Equation (3), we compute $\tau_{geo} \geq \tau_{src}$ for monochromatic emission ($\tau_{dsp} \to 0$) at $E_0$ in the range $1\,\mu m \leq \varnothing_{src} \leq 50\,\mu m$ (FWHM) and with different values for the slope error of the grating substrates, as displayed on the left in Figure 3.

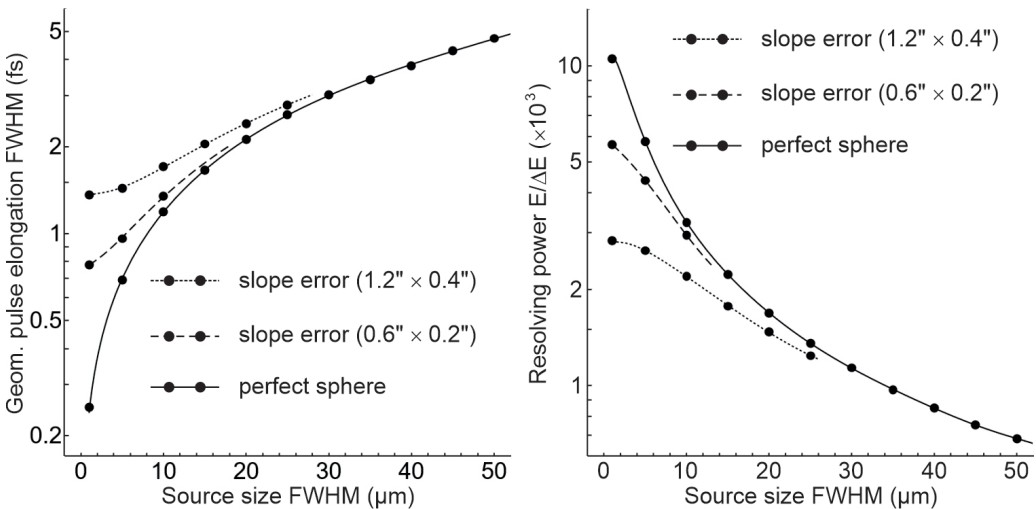

**Figure 3.** Geometrical pulse elongation $\tau_{geo}$ in the sample plane (**left**) and resolving power $E/\Delta E$ (**right**) at the design energy $E_0$ (Table 2) vs. source size for various slope errors (mer. $\times$ sag.) on the RZPs. In the right plot, the diffraction limit for $\Delta\vartheta = 1$ mrad is taken into account via convolution with RT data (black dots). Continuous lines are drawn as a guide for the eyes. Toward large source diameters, the curves coincide asymptotically and overlapping data points are omitted for clarity.

In the presence of surface imperfections on the arcsec scale and the inevitable Fourier limit $\tau_{ftl}$ from Equation (3) for a pulse of finite bandwidth such as, e.g., in Table 1, a source size in the order of $\sim$1 µm would be required to reach the sub-fs regime. Beyond $\varnothing_{src} \approx$ 30 µm, the influence of slope errors on $\tau_{geo}$ is more and more insignificant.

The resolving power $E/\Delta E$ at the design energy $E_0$ is calculated from the spectral dispersion (see Section 5) and the intermediate focal-spot size [7,9] on the PM, as depicted in Figure 3 on the right. $E/\Delta E$ declines with $\varnothing_{src}$, again almost regardless of the substrate quality for a large source but with a notable effect for $\varnothing_{src} \lesssim 10$ µm. In the domain of a point-like source, a spectral diffraction limit of $E/\Delta E$—which equals the number of coherently illuminated grating lines on RZP$_1$ ($\approx 1.15 \times 10^4$) for the given angular divergence—becomes attainable under nearly ideal experimental conditions and in the absence of any misalignment or fabrication inaccuracies. The necessity to minimize or even eliminate slope errors, to exploit the full capacity of the TDCM for small sources, is evident from Figure 3. Potential figure errors of the spherical substrate can be eliminated by customized diffractive wavefront correction (DWC): Established algorithms [27,28] allow for the straightforward and fast computation of the modified grating-line structures of both RZPs, using the phase information either from an ex-situ measurement of the surface profile or in-situ wavefront sensing. Since no further optical elements are needed, the non-invasive and cost-efficient DWC technique paves the natural way to the selection of pulses with a duration well below 1 fs or their monochromatization to $E/\Delta E \gtrsim 5 \times 10^3$, depending on the experimental conditions. On the other hand, many demands on the photon flux can be only fulfilled by an extended source, which we assume to be $\varnothing_{src} = 50$ µm (FWHM) from now on. Even wavefront correction would not significantly improve the performance of real, "wavy" substrates in that case, as shown in Figure 3. Hence, without loss of generality, we restrain all further simulations to the ideal spheres the RZPs are written on.

### 4.2. Pulse Propagation

The capability to cover a broad energy interval with a relatively simple setup represents a key feature of the TDCM as proposed in this research. Indeed, focusing on the PM is accomplished at the moderate expense of resolution for off-design energies ($E_0 \pm 100$ eV) and even in perfect agreement with the theoretical minimum on the intermediate spot size at $E_0$ [9], as depicted in Figure 4.

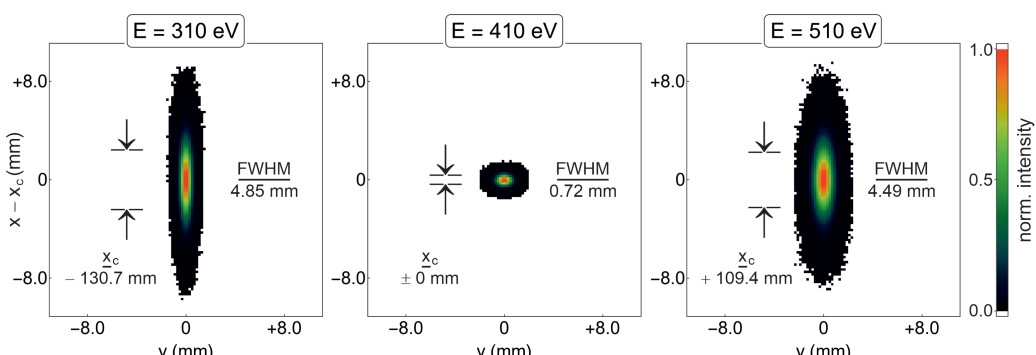

**Figure 4.** Footprint of the pulsed beam on the PM at three photon energies, evaluated from RT simulations for a source size $\varnothing_{\text{src}} = 50\,\mu\text{m}$ and a divergence $\Delta\vartheta = 1\,\text{mrad}$. The meridional width (FWHM) and the central position $x_c$ of the spot on the PM (see Figure 1 and text for details) are noted.

Moreover, the symmetric configuration of the twin RZPs implies 1:1 imaging of the source onto the sample, both in sagittal (H) and meridional (V) dimensions. Nevertheless, residual geometrical aberrations, if evaluated for a Gaussian source distribution ($\varnothing_{\text{src}} = 50\,\mu\text{m}$) at $E_0$ and an open aperture ($\Delta s \rightarrow \infty$) on the PM, blur the focus by about 20% (H) and 5% (V), respectively. Within a relative range of $\pm 7\%$ around $E_0$, that focal spot shape is maintained to a significant extent. Merely at energies far off the design value, the outgoing beam is deformed—like on the PM, but now especially in the sagittal direction, as shown in Figure 5.

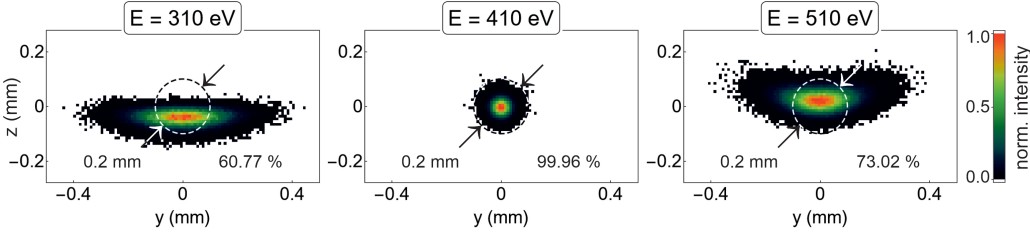

**Figure 5.** Normalized intensity distribution, obtained by Monte Carlo RT, for three photon energies in the sample plane (Figure 1), assuming a source size $\varnothing_{\text{src}} = 50\,\mu\text{m}$ (FWHM). The relative flux (%) inside the dashed circle with a diameter $\varnothing = 0.2\,\text{mm}$ is noted in the lower-right corner of each picture.

As verified by the moderate distortion for 310 eV and 510 eV, the optical scheme as sketched in Figure 1 and defined in Table 2 allows for an operation of the instrument at low chromatic errors, i.e., over a wide spectral range. To provide a well-confined spot on the sample, a pinhole ($\varnothing = 0.2\,\text{mm}$), which still transmits $\sim 2/3$ of the photon flux, is nonetheless inserted in the focal plane.

### 4.3. Transmission Efficiency

To evaluate the optical transmission, as determined by the two RZPs and the PM but without the pinhole, we presume that all apertures are large enough to grasp the full cross-section of the beam, as it is defined by the source parameters. In particular, the reflective stripe on the PM is opened again to avoid any partial blocking of X-rays. In this case, the energy-dependent, relative flux $\mathcal{T}_{\text{tot}}(E)$ through the system can be written as

$$\mathcal{T}_{\text{tot}}(E) = \mathcal{P}^{(2)}_{-1}(E) \cdot \mathcal{R}_0(E) \cdot \mathcal{P}^{(1)}_{+1}(E), \qquad (4)$$

where $\mathcal{P}^{(i)}_{\pm 1}$ denotes the diffraction efficiency of RZP$_i$ with $i \in \{1, 2\}$ in $(\pm 1)^{\text{st}}$ order and $\mathcal{R}_0$ describes the $0^{\text{th}}$ order reflectivity of the PM. In the energy range considered in this work, Ni represents the material of choice for coating the binary ("laminar") grating profile of the RZPs. Using rigorous coupled wave analysis (RCWA, GSolver 5.2™: http://gsolver.com), the depth and relative width of the grooves are optimized to 13.6 nm and 64%, respectively,

for the design configuration (Table 2). An efficiency $\mathcal{P}_{\pm 1}^{(i)}(E_0) = 28.0\%$ is evaluated in case of an ideal, perfectly smooth and clean structure. The real profile is modeled assuming 2 nm of $NiCO_3$ (density $4.4\,\mathrm{g\,cm^{-3}}$) on top of the Ni layer, and $\pm 1$ nm (rms) as an upper limit to the micro-roughness [29]. Including these imperfections, the RZPs contribute an averaged efficiency $\langle \mathcal{P}_{\pm 1}^{(i)} \rangle_E = (25.0 \pm 1.9)\%$ to the overall transmission, whereas the PM reflects $\langle \mathcal{R}_0 \rangle_E = (75.7 \pm 2.1)\%$ across the interval (310–510) eV. By means of the "REFLEC" routine within RAY-UI: https://helmholtz-berlin.de/forschung/oe/wi/optik-strahlrohre/arbeitsgebiete/ray_en.html and considering the slight surface degradation [29] from above, the throughput is computed according to Equation (4) via Monte-Carlo statistics (Table 3).

**Table 3.** Overall transmission, based on diffraction efficiencies of $RZP_{1,2}$ and reflectivity of the PM.

| $E_{\mathrm{phot}} = 310\,\mathrm{eV}$ | $E_{\mathrm{phot}} = 410\,\mathrm{eV}$ | $E_{\mathrm{phot}} = 510\,\mathrm{eV}$ |
|---|---|---|
| $\mathcal{T}_{\mathrm{tot}} = (4.20 \pm 0.06)\%$ | $\mathcal{T}_{\mathrm{tot}} = (6.00 \pm 0.11)\%$ | $\mathcal{T}_{\mathrm{tot}} = (5.33 \pm 0.34)\%$ |

## 5. Resolving Power and Pulse Duration

In comparison to the previous design [18,19] where the TDCM is built of RZPs on plane substrates, the accessible energy range can be considerably enlarged by using RZPs on spheres with an optimal radius of curvature, due to the polychromatic-focusing property of the substrate [20] in terms of a corrected cross-section of the beam in the sample plane (Figure 5). In practice, the transmitted band $E_c - \Delta E/2 \leq E \leq E_c + \Delta E/2$ around a central energy $E_c$ is controlled by the position $x_c$ and width $\Delta s$ of the reflective stripe on the PM, as illustrated in Figure 1. To scan the range (310–510) eV, the stripe needs to be translated between $-130.7$ mm and $+109.4$ mm, relative to its zero position for $E_0$. The slightly nonlinear, concave dispersion [9] $x_c(E_c)$ is characterized by a $2^{\mathrm{nd}}$-order term $d^2 x_c/dE^2 = -2 \times 10^{-3}\,\mathrm{mm/eV^2}$. From the "white" spectrum of an incident pulse with a relatively large bandwidth, the almost uniform reflectivity $\mathcal{R}_0(E)$ of the stripe within $x_c \pm \Delta s/2$ cuts a box-shaped intensity footprint on the PM. However, the assumed normal distribution of the source, both in its lateral shape and angular divergence, leads to a Gaussian energy spectrum of the X-rays that are selected by the stripe as well.

### 5.1. Energy-Resolving Power

With $\Delta E$ as the FWHM of the pulse that passed the pinhole in the sample plane, the resolving power $E/\Delta E$ of the TDCM is calculated in parametric dependence on $\Delta s$, as shown on the left in Figure 6.

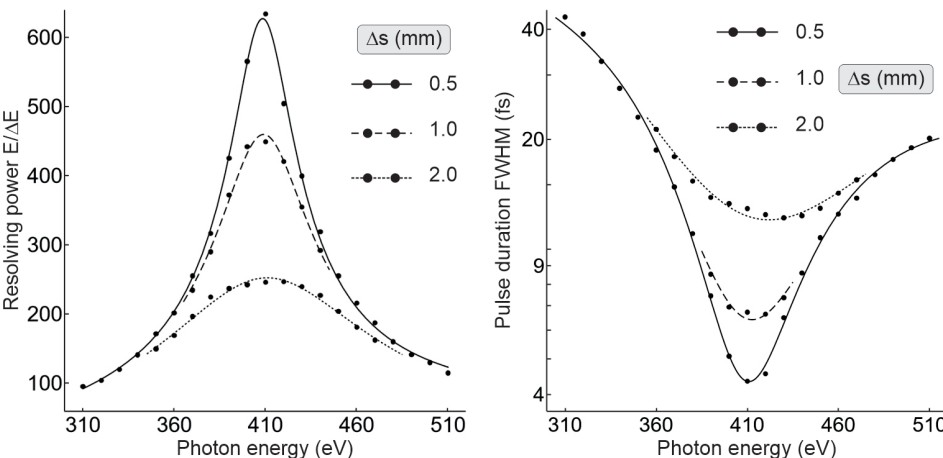

**Figure 6.** Spectral resolving power (**left**) and pulse duration (**right**) of the TDCM as a function of the photon energy at three values for the reflective stripe width $\Delta s$, in each case for a source size of 50 μm (FWHM). Ray-tracing data (dots) follow the nonlinear fit model (solid, dashed and dotted lines) from Equation (5). Toward off-design energies, the curves coincide asymptotically, with reduced plots for clarity.

Statistical fluctuations of the Monte-Carlo data at discrete energies $E_c$ are smoothed by the universal model describing RZP optics [9]:

$$\mathcal{G}_{\Delta s}(E) = \mathcal{G}_l(E) + \mathcal{G}_\star \left[1 + \left[(E - E_\star)/\sigma_\star\right]^2\right]^{-1/2} \quad \text{with} \quad \mathcal{G}_l(E) = g_0 + g_1(E - E_\star), \quad (5)$$

where $\mathcal{G}_\star$, $E_\star$ and $\sigma_\star$ denote peak, central energy and spectral width, respectively, whereas $g_0$ and $g_1$ account for the offset and asymmetry of the fit function $\mathcal{G}_{\Delta s}(E)$. Within the considered parameter range, the maximum resolving power at $E_0$ decreases exponentially with $\Delta s$. On the other hand, the TDCM still provides a resolution of a few eV at distinct off-design energies, even for $\Delta s = 2\,\text{mm}$, enabling a high photon flux over a broad spectral range.

*5.2. Pulse Duration*

The associated lower bound to the duration $\tau_{\text{tot}}$ of the monochromatized pulse is calculated in two steps. First, the component $\tau_{\text{geo}}$ is evaluated via the optical path from the source to the sample, including the effect of far-field diffraction at the stripe ($\sim$10%, depending on $E$ and $\Delta s$). $\tau_{\text{geo}}$ is subsequently convolved with the Fourier limit $\tau_{\text{ftl}}$ due to the finite bandwidth $\Delta E$ provided by the reflective stripe, according to Equation (3). In analogy to $E/\Delta E$, the results of respective Monte-Carlo simulations are displayed on the right in Figure 6. With different parameters, the same fit model from Equation (5) applies to the pulse duration for any choice of $\Delta s$ and again, the sensitivity to the stripe width is highest around $E_0$.

*5.3. Comparison to Fourier Limit*

At this energy and for $\Delta s = 0.5\,\text{mm}$, the TDCM provides a resolution $\Delta E = 0.66\,\text{eV}$. The shortest pulse length $\tau_{\text{tot}} = 4.34\,\text{fs}$ exceeds the Fourier limit of 2.78 fs by 56%. To understand the origin of this elongation, we simulate the same configuration, but now with an ideal point source ($\varnothing_{\text{src}} \to 0$). The resolution is improved to 0.42 eV, with a correspondingly increased value for $\tau_{\text{ftl}}$. However, the geometrical contribution to the pulse duration is the same as for the extended source, $\tau_{\text{geo}} = 3.35\,\text{fs}$, and thus—under given conditions—mainly caused by the dispersion ($\tau_{\text{geo}} \approx \tau_{\text{dsp}}$) along the reflective stripe (of finite width $\Delta s$) on the PM, in agreement with Equation (3). Since $\tau_{\text{ftl}}$ now dominates over $\tau_{\text{geo}}$ for that point source, the—nonetheless enlarged—total pulse duration comes closer to $\tau_{\text{ftl}}$, with an excess $\lesssim 26$%, depending on the specific energy distribution selected by

the stripe. Besides, for narrow stripes, diffraction at its edges gains in importance more and more.

We compare the performance of the TDCM at $E_0$ to the Fourier-transform limit of a pulse whose bandwidth $\Delta E_{ftl}$ just equals the band-pass of the monochromator: For this purpose, the group delay $\mathcal{D}_G(E)$—here expressed as a function of the photon energy for clarity—is evaluated from the simulated arrival times of the pulse train for $(410 \pm 100)$ eV,

$$\mathcal{D}_G(E) \approx \sum_{n=1}^{3} c_n (E - E_0)^n \quad \text{with} \quad c_1 = 7.37 \, \frac{\text{fs}}{\text{eV}}, \, c_2 = -0.02 \, \frac{\text{fs}}{\text{eV}^2}, \, c_3 = 5 \times 10^{-5} \, \frac{\text{fs}}{\text{eV}^3}. \quad (6)$$

The group delay in Equation (6) is an inherent, characteristic function of the optical system and determined by its geometrical parameters (Table 2) but not by the source size. Accordingly, $\mathcal{D}_G(E)$ is evaluated with a "pencil beam," as described in Section 3, to avoid distortions from residual aberrations. With the general formula for $\tau_{ftl}$ (Section 2), the condition $\mathcal{D}_G(E_0 + \Delta E_{ftl}/2) - \mathcal{D}_G(E_0 - \Delta E_{ftl}/2) = \tau_{ftl}(\Delta E_{ftl})$ yields $\Delta E_{ftl} = 0.50$ eV and $\tau_{ftl} = 3.67$ fs. Hence, the not-yet optimized setup from Figure 1 and Table 2 may already operate near the optimal balance between spectral resolution and time response [23].

## 6. Chirped-Pulse Compression

As quantified by Equation (6), the time required by X-rays to propagate through the optical system increases with their energy. Without spectral selection, i.e., an open stripe $\Delta s \to \infty$, that group delay disperses a broadband source into a "train" of quasi-monochromatic, time-delay compensated pulses on the sample.

Vice versa, a chirped source pulse like in Equation (2) is compressed [30,31], if the magnitude and sign of the chirp (in terms of the differential time shift with the instantaneous energy) match the negative group delay dispersion (GDD) $-d\mathcal{D}_G/dE$ of the instrument at least approximately. In other words, a source pulse with a fixed, linear down-chirp of about $-7.37$ fs/eV would be compressed most for $E_c = E_0$ and—due to the non-linearity of $\mathcal{D}_G(E)$—less efficiently at off-design energies. For comparability, we consider pulses at $310 \, \text{eV} \leq E_c \leq 510 \, \text{eV}$ but with the same TBP: The bandwidth is set to $\Delta E = 5$ eV (FWHM) and the corresponding pulse duration of $(37.9 \pm 0.4)$ fs exceeds the Fourier limit by a factor of $10^2$. Results of numerical simulations are displayed in Figure 7.

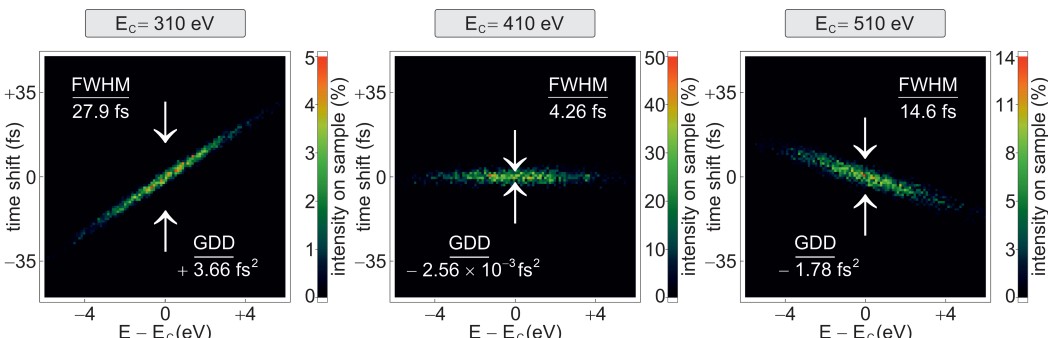

**Figure 7.** Chirped-pulse compression across the range (310–510) eV for a constant time-bandwidth product of the source pulse with a spectral width of 5 eV and a temporal length of $(37.9 \pm 0.4)$ fs. Displayed is the distribution in the sample plane with the pulse duration (FWHM) and the residual GDD in units of $[\text{fs}^2]$. The "intensity on sample" is scaled relative to that of the source, including the transmission efficiency from Table 3. For each Monte-Carlo simulation, $1.5 \times 10^5$ photons are used.

Pulse dimension in the energy-time domain and type (up or down) of the residual chirp are characterized by temporal FWHM and sign of the GDD, respectively. The non-linear group delay from Equation (6) overcompensates the chirp at 310 eV, whereas the compensation is incomplete at 510 eV. Nonetheless, the compression ratio $\mathcal{Q}_C$—defined as the pulse length (FWHM, see Figure 7) of the source divided by that on the sample—is

significantly larger than one over the whole interval of 200 eV. As evaluated from a detailed scan in steps $\delta E_c = 10\,\text{eV}$, the function $\mathcal{Q}_C(E)$ peaks at $E_0$ with $\mathcal{Q}_C = 9.1 \pm 0.6$ (including statistical uncertainties) and declines continuously toward off-design energies, like the resolving power $E/\Delta E$ in Figure 6, for instance. Within a range of 65 eV around $E_0$, at least 50% of that maximal compression is achieved (FWHM), confirming the broadband capabilities of the instrument again. In proportion to $\mathcal{Q}_C$, the intensity of the pulse increases too—and could be many times higher than that of the input signal, in principle. However, in contrast to visual optics, absorption and diffractive loss (Section 4) diminish the gain notably, and the peak intensity on the sample varies between 5% at 310 eV over 50% near $E_0$ to about 14% at 510 eV, as shown in Figure 7.

## 7. Discussion

With the proposed design of a TDCM, we aim to satisfy the demand for an instrument matching the needs in the soft X-ray domain, especially the water window. Research published in the past has often focused on the XUV, where conical diffraction from plane gratings in combination with toroidal [14,23] or—for fewer aberrations—even parabolic [30] mirrors fulfills the requirements of energy resolution, time-delay compensation and a high transmission as well. Since the conical mount could be only realized—if at all—with a poor diffraction efficiency toward the water window and beyond, and the adjustment of six optical elements [14,23,30] is cumbersome, we favor the "two-element" concept [15,18,19] but now with hybrid RZPs that can be fabricated precisely on spheres of a relaxed curvature [20]. Their inherent achromatic capability supports the coverage of a broad spectral range, and the conventional configuration, in which a narrow and sophisticated transmission slit is used to tune the short wavelength range, is replaced by a wide reflective stripe. The tuning is accomplished by one linear translation, rather than rotation of two gratings [15].

In this study, we pick out a specific, albeit realistic design for a proof-of-principle investigation of its optical properties. Nonetheless, parameters such as arm lengths or grazing incidence angles can be adapted to experimental demands. An adjustable group delay is of particular interest, to compress pulses with a variable chirp, and the implications of such variations on the performance must be considered. Different energy bands could be addressed, for instance, by an array of seven parallel RZPs with coatings such as C, Ni, or Au, and the transmission efficiency would range from 9.5% at 100 eV to 0.9% around 1000 eV.

## 8. Conclusions

In conclusion, we demonstrate by simulations the functionality of a newly proposed, versatile time-delay compensated monochromator (TDCM) for ultra-fast pulse shaping over a broad spectral region in the soft X-ray domain. Based on two identical reflection zone plates (RZPs) with technically feasible line densities of $\approx 5 \times 10^2\,\text{mm}^{-1}$ on spherical substrates, our configuration allows for aberration-corrected, flat field detection within a wide energy range up to 0.2 keV. That interval is accessed without rotating parts, and the practical use may further benefit from a relatively simple, mm-sized reflective stripe on an intermediate plane mirror instead of the narrow, expensive transmission slit in the Fourier plane. A smoothly varying transmission efficiency of $(5.2 \pm 0.9)\%$ across the whole spectral range of use supports the calibration of the monochromator or chirp compressor with respect to the photon flux, a demand for the valid interpretation of the measurements.

Around the designed energy of 410 eV, the TDCM is capable of working near its optimum in terms of the spectro-temporal response to a pulse at the Fourier-transform limit [23], whereas a source of 50 μm is assumed for simulations. If the source is reduced to ~1 μm, a pulse length in the sub-fs regime or a resolution well below 0.1 eV could be reached. In its inverse mode of operation, the instrument provides chirped-pulse compression, too, at an intensity level on the sample that is comparable to that of the

incident pulse ($\leq$50%), in spite of the diffractive–reflective loss during the propagation through the optical scheme.

In future, general, preferably analytical rules for the construction of a customized TDCM [23] or "compressor" [30] with hybrid, wavefront-corrected RZPs [26,27] on curved substrates [20] shall be derived. Matching the needs of high harmonic generators [32], laser-produced plasma sources [33] or free-electron lasers (FELs) [21,34] and addressing their characteristics with up- or down-chirps (Section 2), such an optimized instrument may open the door to advanced, even Fourier-limited applications such as time-resolved, ultra-fast resonant inelastic X-ray scattering (RIXS) spectroscopy [35,36]. Methods for the measurement of soft X-ray, (sub-)fs pulses and their parameters should be developed, too.

**Author Contributions:** Conceptualization, A.E.; methodology, C.B. and A.E.; software, C.B. and A.E.; validation, C.B. and A.E; formal analysis, C.B.; investigation, C.B.; writing—original draft preparation, C.B. and A.E.; writing—review and editing, C.B. and A.E.; visualization, C.B.; supervision, A.E.; project administration, A.E.; funding acquisition, A.E. All authors have read and agreed to the published version of the manuscript.

**Funding:** This research, especially the development of hybrid RZPs, was funded by the Berlin Program "ProFiT" in the project "MOSFER" with the grant number 10168769.

**Institutional Review Board Statement:** Not applicable.

**Informed Consent Statement:** Not applicable.

**Data Availability Statement:** The data are available upon request from the authors. All simulations were performed within Mathematica™: https://www.wolfram.com/mathematica/Optica™: https://www.opticasoft.com, the software RAY-UI: https://helmholtz-berlin.de/forschung/oe/wi/optik-strahlrohre/arbeitsgebiete/ray_en.html and the program GSolver™: http://gsolver.com.

**Conflicts of Interest:** The authors declare no conflict of interest. The funders had no role in the design of the study; in the collection, analyses, or interpretation of data; in the writing of the manuscript, or in the decision to publish the results.

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
