# Peer review of "Broadband Time-Delay and Chirp Compensator for X-ray Pulses"

_photonics, doi:10.3390/photonics9050302_

Round 1

Reviewer 1 Report

The time-delay compensated monochromator (TDCM) employing reflection zone plates for the energy range of 310-510 eV is proposed. The proposed system is numerically evaluated.

This is an interesting proposal for a TDCM for the “water window” region. The system is numerically evaluated. One of the interesting results is that sub-femtosecond pulses can be extracted in the case of a sub-micron beam. The system should be useful for spectroscopy in the “water window” region. Therefore, I would like to recommend for publication in Photonics, when the following point is clarified:

The authors discuss the chirp compensation. In the TDCM, each spectral component separated by the zone plate propagates noncollinear and is recombined by the second zone plate. Because the propagation direction of each spectral component is determined by diffraction, the propagation direction cannot be adjusted independently. Therefore, in my opinion, if the temporal chirp is compensated, the spatial recombination of the beam becomes incomplete and therefore the beam has a spatial chirp. Strictly speaking, the TDCM is misaligned. What is the tolerance of the alignment of the TDCM?

Reviewer 2 Report

The authors present an interesting proposal for an optical concept to monochromatize soft X-rays close to the Fourier limit in a time delay compensating desing using two diffractive elements with energy bandwidth selection in between. By using reflection zone plates on spherical substrates, the authors improve upon a design that they proposed earlier. While the overall paper is well-written and clearly presented, I have some minor points for potential improvements:

1. Around line 42, the authors intorduce the time-bandwidth product (TBP) as the Fourier transform limit for Gaussian pulses. For the non-specialist reader, it would be helpful to mention that the TBP depends on the actual pulse shape and different pulse shapes lead to different TBP values (e.g. a sech^2 pulse shape yields smaller values). Stating the actual value of the TBP would also be instructive.

2. In lines 47-48, the authors claim that "to access the sub-femtosecond regime, and ~44 oscillations of the electric field are passed within \tau_ftl, independent of the energy." This statement is wrong, the actual number of oscillations of the electric field (as well as the relative bandwidth stated in the line before) linearly depends on the energy. The stated value may be roughly correct within the considered energy range, but does not apply in general.

3. Around line 83, the authors introduce the concept of using a reflective stripe of a plane mirror (ideally of variable width) in order to select chosen energy bandwidths. A few statements should be added:
a) How can a variable width of this strip be realized in practice,
b) How to prevent scattering from the edges and the area outside of the reflective strip, and
c) What is the influence of edge diffraction from the strip? 

4. After line 176, formula (5) is introduced. The addition of G_l as a sum over n with n running from 0 to 1 seems overly complicated, since the term with n=0 only adds the constance g_0. This could be displayed in a simpler fashion to not confuse the reader.

5. After line 194, formula (6) is introduced to quantify the group delay that is introduced by the proposed device. To my understanding, it is shown that for a certain configuration using a certain parameter set, especially a fixed source size, the geometrical contributions dominate the introduced group delay. The authors show that for a specific parameter set, operation very close to the Fourier limit is achieved. In the following section, the authors continue to discuss how the introduced group delay can in fact be used to compress a pre-chirped pulse. It remains unclear, how fixed the stated value of the introduced group delay is and how it depends on the chosen parameter set (is it really dominated by the source size?).
To be useful as a compressor, a tunable group delay would be desirable. The authors should explain if and how the group delay can be altered and what impact that has on the other performance parameters of the device (i.e. transmission, resolution and so on).

Reviewer 3 Report

The article by Christoph Braig and Alexei Erko, titled Broadband Time-Delay and Chirp Compensator for X-ray Pulses, describe a modification of a precedent design by the same authors.

The article proposes the use of spherical substrates and a flat mirror in spite of the exit slit as the major modifications.  The idea seems valid but some technical details are missing and some assumptions are not necessarily a technical advantage.

It is unclear how a reflecting mirror can be easier to handle than a simple and very common slit. Intuition would say exactly the contrary. There are no details on the precision of the translation of the mirror and the need of preserving the angle of incidence so, it's hard to judge how easy is to move this mirror but, surely, seems less simple than a simple slit.

Also, the radius of curvature is not presented. If the radius is short, how easy is to fabbricate the RZP on an highly curved surface? The authors are experts so, they can judge the easiness but, the reader may be not. So, it would be important to add the value of the ROC for this design. Also, is the eventual path difference, due to a non planar surface, being taken into account to calculate the induced time stretching? 

The statement that a customized wavefront corrector can eliminate the aberrations is, in principle, correct. But it is something invasive, costly and cumbersome to use so, the aberrations should be taken into account, especially if ROC is small and the figure errors and spherical aberration can be significant. 

Some other minor comments are:

Line 7 (abstract). The word "respectively" here seems out of contest. Please check. 

Line 83: Note sure if the reference is the appropriate here for explaining the geometrical elongation.

Table 2: R i-j value(s) should be written in figure 1 and not in this table. 

Line 98: I understand what the sentence "In the case of a holographic diffractive structure" means but, it my not be clear to a general reader. Please explain better. 

Line 190: An exceeding of 50% WRT the TL, seems quite a large value for a compensated monochromator. What are the limiting factor (the general statement that is the geometrical effect may not be too explanatory)

Line 231: Stating that "even for a large source of 50 μm" is not correct, I think. If the source is larger, shouldn't it be easier to stay closer to the transform limit?

Line 244: Not sure one can say that this modified design can provide diffraction limited spot. Nothing here address the wavefront preservation to achieve the proper Strehl Ratio so, I would remove any reference to diffraction limited spots.

Reviewer 4 Report

The authors propose a new type of monochromator for soft X-ray FEL pulses, where two reflection zone plates and flat mirror is included. I think this concept is important for the field of XFEL and is suitable to publish in Photonics. I have one minor question about the energy range of monochromator. In Figure 1, the authors describe that the design energy can be changed by the position of plane mirror (Xc). The design energy is set to be 400 eV. How much is a minimum and maximum of the design energy in this monochromator system?

Round 2

Reviewer 1 Report

The manuscript was revised to answer my question. So, I recommend it for publication in Photonics.

Reviewer 3 Report

Thanks for replying to all my comment. I think the article is perfectly suitable for publication and contains interesting aspect for the interested readers.